# Detection and Characterization of TiO_2_ Nanomaterials in Sludge from Wastewater Treatment Plants of Chihuahua State, Mexico

**DOI:** 10.3390/nano12050744

**Published:** 2022-02-23

**Authors:** Juan Reyes-Herrera, Damaris Acosta-Slane, Hiram Castillo-Michel, Ana E. Pradas del Real, Katarina Vogel-Mikus, Federico Benetti, Marco Roman, Julie Villanova, M. Cecilia Valles-Aragón

**Affiliations:** 1European Synchrotron Radiation Facility, B.P.220, CEDEX 09, 38043 Grenoble, France; juan.reyes-herrera@esrf.fr (J.R.-H.); hiram.castillo_michel@esrf.fr (H.C.-M.); julie.villanova@esrf.fr (J.V.); 2Faculty of Agrotechnological Sciences, Autonomous University of Chihuahua, Campus 1, Pascual Orozco, Chihuahua 31350, Mexico; dslane@uach.mx; 3Department Agroenvironmental Research, Madrid’s Institute for Rural Research and Development, Agricultural and Food, Leganitos 47, 28013 Madrid, Spain; ana.elena.pradas@madrid.org; 4Department of Biology, University of Ljubljana, Večna pot 111, SI-1000 Ljubljana, Slovenia; Katarina.VogelMikus@bf.uni-lj.si; 5Jozef Stefan Institute, Jamova 39, SI-1000 Ljubljana, Slovenia; 6EcamRicert Srl, European Centre for the Sustainable Impact of Nanotechnology, Corso Stati Uniti 4, 35127 Padua, Italy; f.benetti@ecamricert.com; 7Department of Environmental Sciences Informatics and Statistics, University Ca’ Foscari, Dorsoduro 2137, 30123 Venezia, Italy; marco.roman@unive.it

**Keywords:** nanoparticles, sewage sludge, X-ray microspectroscopy, synchrotron

## Abstract

TiO_2_ nanoparticles (TiO_2_-NPs) have a wide range of industrial applications (paintings, sunscreens, food and cosmetics) and is one of the most intensively used nanomaterials worldwide. Leaching from commercial products TiO_2_-NPs are predicted to significantly accumulate in wastewater sludges, which are then often used as soil amendment. In this work, sludge samples from four wastewater treatment plants of the Chihuahua State in Mexico were obtained during spring and summer (2017). A comprehensive characterization study was performed by X-ray based (laboratory and synchrotron) techniques and electron microscopy. Ti was detected in all sludge samples (1810–2760 mg/kg) mainly as TiO_2_ particles ranging from 40 nm up to hundreds of nm. Micro-XANES data was analyzed by principal component analysis and linear combination fitting enabling the identification of three predominant Ti species: anatase, rutile and ilmenite. Micro-XANES from the smaller Ti particles was predominantly anatase (68% + 32% rutile), suggesting these TiO_2_-NPs originate from paintings and cosmetics. TEM imaging confirmed the presence of nanoscale Ti with smooth surface morphologies resembling engineered TiO_2_-NPs. The size and crystalline phase of TiO_2_-NPs in the sludge from this region suggest increased reactivity and potential toxicity to agro-systems. Further studies should be dedicated to evaluating this.

## 1. Introduction

### 1.1. NanoTiO_2_

Due to their novel properties, engineered nanomaterials (ENMs) are extensively used in many commercial products [1], processes and industrial applications [2]. TiO_2_ nanoparticles (TiO_2_-NPs) are one of the most produced ENMs worldwide [3], with an estimated production of 88,000 metric tons/year in 2010 [4], and a forecast of 2.5 million tons by 2025 [5]. TiO_2_-NPs are used in many applications/areas, including food (food-grade TiO_2_/E171), cosmetics, personal care (sunscreen, toothpaste and deodorants) [6], agriculture, medicine, environmental protection [7], and engineering (solar cells [8] and building materials [9]). These NPs are particularly used in paints and coatings to provide surface self-cleaning and self-disinfecting properties [10]. Despite the benefits of using ENMs, there are also several concerns about their fate and environmental implications [11,12,13].

### 1.2. TiO_2_-NPs Release to the Environment

During their manufacturing, transport, usage or disposal, ENMs are unavoidably released into soil, water or air [14]. The latest dynamical models estimating the emissions of ENMs into the environment, agree that TiO_2_-NPs are amongst those with the highest mass-flow release [15,16], other studies indicate that sewage is the main release pathway of TiO_2_-NPs [1,17]. TiO_2_-NPs may release from paints on building facades and reach urban runoff, in addition TiO_2_-NPs from food are released by human excretion, and TiO_2_-NPs from cosmetics and personal care products are washed and disposed into sewage systems [18]. Other models also predict a significant release into freshwater, urban and agricultural soil (where biosolids or treated water are the main sources) [19]. On the other hand, the Organization for Economic Cooperation and Development considers sludge treatment for sewage as the most used method in the waste-water treatment plants (WWTPs), worldwide [20]. Studies show that during a treatment process in a WWTP, more than 90% of ENMs are retained in the sludge [21], up to 92% for TiO_2_-NPs specifically [22]. Therefore, sludge from WWTPs can be considered as the main accumulator of TiO_2_-NPs coming from anthropogenic activities, as it has been pointed out by many authors [23,24].

### 1.3. Fate of TiO_2_-NPs in WWTPs Sludge

A number of studies on characterization of TiO_2_-NPs in full scale WWTPs detected these nanoparticles in sludge or wastewater. Kiser et al. [18] reported concentrations of 1 to 6 mg Ti/kg in one WWTP located in Arizona (U.S.A.). The same researchers detected TiO_2_-NPs in wastewater effluents from 10 municipal facilities from southern to central Arizona. Measuring concentrations from <2 to 20 μg/L, they estimated that on average 98.3% of the incoming Ti is removed by the wastewater treatment [25]. Kim et al. [26] reported a total Ti concentration in three sewage sludge products from the US EPA Targeted National Sewage Sludge Survey ranging from 96.9 to 4510 mg/kg, in which they identify TiO_2_ particles from 40 to 300 nm. Tong et al. [27] found direct evidence of the presence of TiO_2_ in sludge from a WWTP in Skokie, Illinois (U.S.A.), in particular with coexistence of both anatase and rutile phases with particle size in the range 100–300 nm. Shi et al. [28] investigated the fate of TiO_2_-NPs in two full scale WWTPs in northern China, they found Ti in the effluent from both plants and their receptor streams, predominantly in the form of TiO_2_-NPs agglomerates. In sewage sludge samples collected from 26 WWTPs in Shanghai, China, Tou et al. [29] found anatase and rutile aggregates with sizes ranging from 50 to 400 nm. Polesel et al. [30] studied two WWTPs in the city of Trondheim (Norway), and observed agglomerates and aggregates of TiO_2_ particles from 500 nm to >1 μm in size. Choi et al. [31] conducted a full year study in a WWTP in Maryland (U.S.A.). They found inflow TiO_2_ concentrations in the range 21.6 ± 5.0 to 391.0 ± 43.0 μg/L and estimated a yearly average capture of TiO_2_ of 39.8 and 25.1 kg/day by the primary and the secondary sludge particulates, respectively. Wielinski et al. [32] reported concentrations of 1000–1800 mg Ti/kg in sludge samples from a municipal WWTP in Wallisellen, Switzerland, where anatase and rutile were identified as the main mineral bearing species. By spiking these sludge samples with anatase or rutile NPs their transformations due to the incineration process were determined and both phases were susceptible to transform into members of the hematite-ilmenite solid solution. The analytical techniques used in the above-mentioned studies are summarized in Table A1 of Appendix A.

The worldwide sewage sludge production is considerably increasing, and its management represents an important environmental issue. In the EU-27, the production of dry WWTP sludge is estimated to be around 10.13 MT/year, and it was expected to increase to 13 MT/year by 2020 [33]. In 2017, Seiple et al. reported that 13.84 MT/year of dry solids are captured during wastewater treatment in the U.S.A. [34]. The organic matter and nutrients content in sludge may improve the physical, chemical, and biological properties of soils for crop development [35]. According to the report of European Commission published in 2010, 39% of sewage sludge produced in the EU is recycled through agriculture [36]. Meanwhile, in the U.S.A. the application of biosolids reaches 22.4 dry metric Ton/ha [37]. Current regulations define permissible levels for heavy metals, weeds, human and plant pathogens [38] for the reuse of sewage sludge in agriculture. Currently, Ti content (bulk or nano) is not considered by any regulations worldwide [39,40,41].

### 1.4. Potential Risks of TiO_2_-NPs

Once the sludge from WWTPs is applied as soil amendment, the potential eco-toxic effects of TiO_2_-NPs remain unknown because the mechanisms of interaction between such NPs and natural systems are still to be elucidated [42]. However, a relevant toxicity risk arising from the management and transport of sludge material should not be excluded. The National Institute for Occupational Safety and Health (NIOSH) indicated TiO_2_-NPs as a potential carcinogen and recommended exposure limits of 2.4 for fine and 0.3 mg/m^3^ for ultrafine TiO_2_ (including engineered nanoscale), as time-weighted average concentrations for up to 10 h/day during a 40 h-work week [43]. Furthermore, a recent review on the effects of TiO_2_-NPs exposure based on in vitro and in vivo murine studies, concluded that [7]: intestinal mucosa, brain, heart and other internal organs can be affected by a constant exposure to TiO_2_-NPs at minor doses, which may lead to increased risks of developing several diseases, tumors or acceleration of existing cancer progression.

As mentioned above, the effects of TiO_2_-NPs in soils and biological systems such as plants, crops, and rhizosphere microorganisms are still uncertain. For instance, Simonin et al. [44] found that TiO_2_-NPs trigger cascading negative effects on denitrification enzyme activity and a substantial modification of the bacterial community structure. On the contrary, Moll et al. [45] did not observe negative effects of TiO_2_-NPs on soil microbial communities, while Timmusk et al. [46] showed that TiO_2_-NPs can even enhance the activity of growth-promoting rhizobacteria.

Even if many studies investigated the interaction of TiO_2_-NPs with plants, most of them were performed in hydroponic conditions, whereas just a few considered more realistic environments. Amongst the latter, cucumbers growing in soils spiked with TiO_2_-NPs were tested by Servin et al. [47], who reported translocation of TiO_2_-NPs from the soil to the edible part of the plants. Their findings suggest that these NPs could enter the food chain, with unknown consequences on the human health. Giorgetti et al. [48] grew peas in farming soil amended with biosolids containing TiO_2_-NPs. They found different damages in the plants, including in the root cells. However, Bakshi et al. [49] reported no significant Ti enrichment in tomato fruits due to sewage sludge amendment containing TiO_2_-NPs, finding no evidence of acute toxicity on plants.

Overall, literature results are inconsistent and there is no consensus on the environmental effects of TiO_2_-NPs, but it is evident that chemical speciation and particle size plays a crucial role in determining the toxicological properties of TiO_2_-NPs [50,51,52,53]. Nonetheless, the physicochemical characterization of ENMs in natural media is challenging due to the matrix complexity and the rather low concentrations [42,52].

This study presents a comprehensive physicochemical characterization of TiO_2_-NPs in sludges collected from the four existent WWTPs in the Chihuahua State, Mexico. Three WWTPs process the waste of the two main cities in the State, serving a population of around 1 million; the fourth locates in a less industrialized small city, therefore a lower concentration of TiO_2_-NPs is expected. All the sludge from all plants is used to amend nearby fields, representing a potential environmental risk. Therefore, assessing the presence and characteristics of TiO_2_-NPs in the WWPTs sludge is of primary importance. A variety of complementary analytical techniques were used in this study, starting from bulk techniques: bulk X-ray Fluorescence (XRF) spectrometry, Ti K-edge bulk X-ray Absorption Near-Edge Structure spectrometry (XANES), bulk X-ray Diffraction (XRD); to microscopy techniques: micro-XRD, micro-XANES, micro-XRF and nano-XRF, including transmission electron microscopy coupled to energy dispersive X-ray spectroscopy (TEM-EDX). To the best of the authors’ knowledge there are no other studies performed in WWTPs from Mexico, where the sludge is intensively used as agricultural amendment.

## 2. Materials and Methods

### 2.1. Sampling and Sample Preparation

During spring (P) and summer (S) of 2017, sludge samples were collected from four WWTPs of the Chihuahua State in Mexico [54], Figure 1. Two WWTPs were located in the city of Chihuahua, at North (CN) (430 L/s effluent) and South (CS) (1620 L/s effluent), with a sludge generation of 65,700 m^3^/year [55] in both sites. A third WWTP is located in the city of Juarez (JZ) (1620 south effluent, 1350 L/s effluent). Chihuahua and Juarez are the biggest cities of the State with an overall sludge generation of 85,116 m^3^/year [56]. The fourth WWTP locates in Casas Grandes (CG) (100 L/s effluent, sludge production data are unavailable), a smaller city with primary activities. The sampling was performed following the official Mexican normative NOM-004-SEMARNAT-2002 [39]. The acronyms of the cities reported above, joined with the initial of the season, were used to name the samples, e.g., Juarez Spring is JZP. The collected sludge was taken in a semisolid form at the dump of the anaerobic digestor, which is fed by the biological reactor. For hygiene reasons and better preservation of the sample, the sludge was dried at ambient temperature and stored in plastic bags with hermetic closure. Then, pressed pellets of approx. 1 g were prepared from the sludge. In addition, water extracts for each site were prepared from 0.2 g of powdered sludge mixed with 1 mL of deionized water, hand-shaken for one minute, left settling for 30 min and decanted. In this case, 10 μL of supernatant were finally deposited onto Ultralene^TM^ film.

### 2.2. Bulk Analysis

Bulk XRF measurements were performed on all sludge samples pressed in pellets. Each sample was measured twice with a portable XRF spectrometer, Peduzo P01 (Jozef Stefan Institute, Slovenia) equipped with Rh X-ray tube (30 kV) and SDD (Amptek, Inc.; Bedford, MA, USA) detector [57]. The quantitative analysis was performed by the Quantitative X-ray analysis system software package developed in Lab View. The averaged values have an associated uncertainty that includes the statistical uncertainty of measured intensities and the uncertainty of the mathematical fitting procedure [58]. Then, the reported uncertainty corresponds to the combined standard uncertainty. GeoPT standard reference material OU-10 Longmyndian Greywacke was used to validate the quantification procedure. 

Samples from both seasons and all WWTPs were used for bulk XRD. Agatha mortar ground sludge was introduced in thin glass capillaries and mounted in the high energy resolution powder diffractometer at ID22 [59] of the European Synchrotron Radiation Facility (ESRF) in Grenoble. Using a 29 keV incident photon beam, the diffracted X-rays were collected by the multianalyzer stage.

Ti K-edge bulk XANES measurements were performed on the sludge pressed pellets, using a scanning x-ray microscope at the ID21 beamline, ESRF [60]. One measurement was acquired per sample using unfocused mode beam (250 μm diameter); with an average flux of 4 × 10^9^ photons/s. X-ray emission was integrated by a silicon drift detector (SDD, SGX sensortech 80 mm^2^ active area) and the incoming beam flux measured by a photodiode. All energy scans were performed from 4.95 keV to 5.1 keV, with 0.5 eV energy steps with 100 ms dwell time per energy step. Spectra energy was calibrated using a Ti foil reference.

### 2.3. Microscopy Analyses

Both pressed pellets and water extracts were employed to perform coupled μXRF and μXANES measurements using a scanning X-ray microscope at beamline ID21, ESRF. Kirkpatrick-Baez focused beam of 0.8 × 0.4 μm^2^ (HxV) and an average flux of 1.2 × 10^10^ photons/s was used, and energy selection carried out by means of Si (111) double crystal monochromator. XRF emission spectra were obtained with a SDD of 80 mm^2^ of active area (SGX Sensortech). From Ti distribution in the sample, points of interest were selected for Ti K-edge μXANES scans from 4.95 to 5.1 keV (in fluorescence mode). Depending on the size of the Ti hotspot, XANES spectra were labeled as Large for particles with a diameter larger than 3 μm, or Small for particles with smaller diameter. Full X-ray Absorption Spectroscopy (FXAS) mapping mode [61] was used for the water extracts. In this latter case, μXANES spectra were extracted from a stack of images recorded at each energy step of the spectrum, with a step size of 0.5 μm (horizontal) × 0.5 μm (vertical). Hundreds of XRF images were recorded using a region of interest selective for Ti Kα emission lines, corrected for the detector deadtime and normalized by the incoming photon flux measurement (I_0_), aligned using Elastix [62] and saved to a.hdf5 file containing the intensity and energy values for each image. μXANES spectra were then extracted from the FXAS maps using PyMCA [63]. 

Nano-XRF maps were obtained at the beamline ID16B, ESRF [64], at 17.5 keV (ΔE/E = 10^−2^) with a 60 × 60 nm^2^ spot size and an average flux of 2 × 10^10^ photons/s on samples from CNS, JZS and CGS (including water extracts from JZS and CGS). Fluorescence emission spectra were acquired by a VORTEX single element silicon drift detector (SDD) and a three-element SDD array from SGX Sensortech. Particle size distribution was obtained using image segmentation analysis on the Ti-nanoXRF maps. The image processing method is described in supplementary information of Pradas del Real et al. [24].

For the TEM analysis, an aliquot of 5 g of the sludge sample was mechanically ground by mortar and pestle for 5–10 min, depending on the grain size of different samples. Once homogenized, an amount of 3 mg of ground powder was weighed and a volume 1.5 mL ultra-pure water was added. Suspensions were tip-sonicated using a QSONICA sonicator. The suspensions were diluted to acquire a transparent solution, normally 1:100. Three µL of suspension were deposited onto standard electron microscopy grids 400 mesh copper (TED PELLA), previously pre-treated with alcian blue, and let dry.

TEM images were acquired using a 200 KV, JEM-2100 plus electron microscope with lanthanum hexaboride crystal (LaB_6_) gun equipped with a 12 Mpix NanoSprint1200 camera. Dark field images were taken by JEOL STEM detector (Jeol Ltd., Tokyo, Japan) and EDS analysis were extracted by 30 mm^2^ JEOL silicon drift detector (JED-2300T, Jeol Ltd., Tokyo, Japan).

### 2.4. Statistical Analysis

The spectra of all XRF (micro and nano) maps from experiments performed at the ESRF were fitted by PyMCA. Principal component analysis (PCA) was applied on all μXANES data, performed with Orange software [65], including the add-on Spectroscopy [66]. PCA was implemented on the second derivative of the spectra (using the Savitzky-Golay method, second polynomial order and 15 points window) and a vector normalization. From this analysis, three main groups were obtained and the average XANES spectra was then submitted to a linear combination fitting procedure in ATHENA software [67]. Fittings were performed with combinations of three model compounds, anatase and rutile, from Sigma Aldrich (Inc., St. Louis, MO, USA) measured at ID21, ESRF, and ilmenite spectrum obtained from the XAS database of the beamline 10.3.2 at the Advanced Light Source, Berkeley, CA, USA.

## 3. Results

### 3.1. Bulk X-ray Analyses

The total concentrations of Ti in sewage sludge are shown in Figure 2. All sites presented Ti content above 2 g/kg, with no significant difference (within the experimental uncertainty) between seasons except for CSS. Additionally obtained from XRF analysis, multi-elemental concentrations in WWTP sludge are presented in Table 1. 

The crystalline forms of TiO_2_ present in the sludge samples were investigated by synchrotron based High Resolution XRD. Results shown in Figure 3 indicate the most intense diffraction peaks, belonging to anatase and rutile forms. The characteristic diffraction peaks of anatase and rutile isoforms of TiO_2_ are shown in Figure A1. 

Another important insight of the presence of TiO_2_ species in the samples was obtained by bulk Ti K-edge XANES measurements, whose spectra are shown in Figure 4. The spectra from the sites and seasons present spectral features similar to the anatase reference. These results point towards anatase isoform being the predominant species, but, as observed from the XRD data rutile is also present in these samples.

### 3.2. Microscopy Analysis

As mentioned in Section 2.3, after obtaining a map of total Ti distribution, Ti K-edge μXANES spectra were acquired from rich-dense Ti spots, an example from JZP is shown in Figure 5. Each Ti K-edge μXANES spectrum was labelled as Small (red spots) if originated from a spot smaller or equal to 3 μm in size, or Large (blue spots) otherwise.

FXAS measurements from water extracts were included as well, where all spectra are labelled as Small. An equivalent number of Ti K-edge μXANES spectra were collected from all samples, seasons, and size classes, resulting in 550 spectra (see Figure A2), then pooled for principal component analysis (PCA). Three main groups were identified in the PCA transformed space (Figure 6a), whose average spectra are shown in Figure 6b.

While two thirds (67%) of all analyzed particles are Small and one third are Large, the size distribution differs considerably amongst the PCA-defined spectral groups. Indeed, Group 1 and Group 2 are mainly formed by Large particles. Group 3 represents 83% of the overall population and contains 98% of all Small particles. It is important to mention that no relationship was found between TiO_2_ isoforms and city, or season. These two categories did not aid the interpretation of the PCA plot (Figure 6). 

To investigate the particle size of the Ti phases, present in the sludge, a sub-set of samples was investigated using nanoXRF, CGP, JZP and CNP (Figure 7, left side). Due to the limited access to the nanoXRF beamline ID16B, the sub-set of samples was chosen based on the evidence that the cities and seasons do not present significant differences in Ti content, bulk-XRD and bulk-XANES speciation. From the obtained nanoXRF maps, image segmentation analysis was performed to acquire the particles diameter distribution (Figure 7, right side). Different particle sizes were found where around 50% are below 240 nm for the three samples, CGP stands out with 41% smaller than 120 nm.

A few nanoXRF maps were also obtained from water extracts from JZP and CGP (Figure A3), showing the presence of Ti-NPs, where Zn and Cu nanoparticles where also identified.

Based on TEM analyses, sludge samples contained Ti-NPs with a typical diameter ranging from a minimum of 40 nm to a maximum of a few hundreds of nm (Figure 8). The Energy-dispersive X-ray (EDX) spectra indicate that particles contain Ti and O only (Figure A4). 

## 4. Discussion

As the environmental effects of TiO_2_-NPs originated from anthropogenic activities are still under research [42], total Ti concentration is currently not regulated in biosolids in Mexico as much as in other countries worldwide [39,40,41]. In the sludge samples analyzed throughout this study, total Ti concentrations were consistent with other field studies (Table 2).

Sewage sludge analysis for the total concentration of trace elements in all WWTPs and both seasons demonstrated low risk of exposure to Pb (<300 mg/kg), Cr (<120 mg/kg), Cu (<1500 mg/kg), Ni (<420 mg/kg) and Zn (<2800 mg/kg) (Table 1), in compliance to Mexican regulation NOM-004-SEMARNAT-2002 [39]. In addition, the elemental concentrations were in compliance to most international regulations [63,64]. The high levels of some essential elements in the sludge of all sampled WWTP, including P (>8450 mg/kg), K (>1065 mg/kg) and Ca (>18,000 mg/kg) (Table 1), their high volume of production and easy availability at affordable cost, justifies its common use as an agricultural amendment in the nearby agricultural fields [33]. 

High-resolution XRD analyses of the sludge have shown some diffraction peaks typical of anatase and rutile (Figure 3). Unfortunately, some rutile characteristic diffraction peaks were not identified. This is due to overlapping peaks from other components present in the sludge. For instance, in Figure 3c the rutile peak (101) is indistinguishable. Thus, these experiments were followed by XANES, a more selective technique for Ti speciation sensitive enough to distinguish the TiO_2_ isoforms. Since the energy of the X-ray beam is tuned around the Ti K-edge, interference from other elements is very low [70]. A clear signature of TiO_2_ was obtained from bulk Ti K-edge XANES (see Figure 4), the spectra from all sites and both seasons’ samples have spectral features similar to the anatase reference compound. However, from these data it was not possible to infer the TiO_2_ particles size. Actually, this concern was remarked in the study by Tong et al. [27], where bulk Ti K-edge XANES was also used for WWTPs sludge characterization.

Thus, micro-XANES was used for the characterization of TiO_2_ at the microscale. Once Ti was detected in the samples by micro-XRF mapping, micro-XANES spectra were acquired in the hotspots of the element, labeling each spectrum based on the spot size, looking for a possible chemical speciation dependence on size. After an extended Ti K-edge μXANES measurement campaign, and using PCA analysis, three main groups of hotspots were identified (Figure 6): two main TiO_2_ species were associated with large particles and one with small particles (<3 μm). These groups resemble the most common TiO_2_ phases, i.e., rutile, ilmenite and anatase (Group 1, 2 and 3, respectively); see Figure 9.

While the spectra of Group 1 and Group 2 were substantially matching those of rutile and ilmenite references, respectively; Group 3 (which contains 83% of all measurements and consists of all small particles by 98%) resembles anatase, except for some minor differences. The main spectral difference between Group 3 and anatase is observed just after the sharp intense peak due to the rising absorption edge, referred as “white line”, which may indicate a minor contribution of rutile. Linear combination fitting (LCF) was performed on the group spectra and confirmed the predominance of rutile (81% + 19% ilmenite) in Group1, ilmenite (77% + 23% rutile) in Group 2, and anatase (68% + 32% rutile) in Group 2 (see Figure A5). These results are in agreement with those reported by Niltharach et al. [71] that obtained similar Ti K-edge XANES spectra from TiO_2_ nanostructures having high content of anatase and minor content of rutile. Anatase was the predominant phase for smaller particles in the sludge from all sites in our study. In contrast, Tong et al. [27] determined rutile as the dominant phase in their WWTP sludge samples, with 60 ± 3% rutile, 31 ± 2% anatase, and 11 ± 3% ilmenite. The µXANES results obtained in this study suggest that nano-anatase containing products are the main source of Ti. This isoform is used in pigments, indoor paintings and coatings due to its photocatalytic properties, as well as food additive [72]. Rutile was only a minor phase identified in all samples. This TiO_2_ isoform is mainly employed in UV light stabilizers and in sunscreens as a nanocomposite with sub-micrometric size [72]. Here it is important to mark that μXANES imaging not only allows to determine the spatial distribution of TiO_2_ in the sample, but also increased the capability to resolve its isoforms; in comparison to bulk XANES (Figure 4), where the dominating anatase isoform, masks rutile and ilmenite signals. 

NanoXRF measurements showed clear evidence of Ti-NPs presence in all the samples, with characteristic particle size distributions (Figure 7). In all cases, more than 50% of the Ti-NPs present were less than 240 nm in diameter. These dimensions were confirmed by TEM where particle sizes varying from 40 nm to hundreds of nm were measured. It is important to emphasize the scope of the nanoXRF technique, which was able to give a wide perspective of distribution and identification of NPs in almost intact sludge samples (only pelletized, avoiding any kind of liquid extraction). However, some of the identified Ti particles could be clusters of smaller particles, as TEM images show (Figure 8). The particle size distributions found in this study are consistent with previous observation by Kim et al. [26], Kiser et al. [18] and Tong et al. [27]. Additionally, nanoXRF mapping in the water extracts revealed the presence of Cu and Zn. Furthermore, Ti detection after basic water extraction for TEM analysis showed that the element is easily mobilized in the sludge matrix. Mobility of TiO_2_-NPs is crucial for their environmental fate and behaviour; in soils increased mobility increases the risk of transfer into the food chain [73].

TEM images (Figure 8) showed that Ti-NPs in sludge have regular shapes and smooth surfaces. In a characterization research of the natural colloidal Ti background in soil, Philippe et al. [74] found that morphologies and sizes of natural Ti-NPs may vary widely, thus limiting our capability to discern natural and engineered nanoparticles based on morphology. However, natural Ti-NPs present rough surfaces, whereas engineered Ti-NPs present smoother surfaces, as it has been pointed out by Pradas et al. [24] based on TEM images. In this study, Ti-NPs mainly present smooth surfaces, which actually resemble engineered TiO_2_-NPs, such as P25 used in the study of Nickel et al. [75]. P25 are one of the most efficient commercial photo-catalyst TiO_2_-NPs, used in the industry of photo catalysis and in cosmetics, constituted by anatase (82–86%) and rutile (18–14%) [60,63]. Furthermore, higher presence of anatase in sludge samples supports the hypothesis of its artificial origin [72] because in nature the most common minerals of Ti are rather ilmenite, rutile, brookite, and more rarely anatase [76]. 

## 5. Conclusions and Future Perspectives

This study has provided evidence of the presence of nanoscale TiO_2_ in sludge samples from four WWTP from Chihuahua, Mexico collected in spring and summer 2017. The TiO_2_-NPs range in size from 40 to few hundreds nm. TiO_2_-NPs content did not significantly differ from sampling site or season. Based on a large data set of micro-XANES three main Ti phases were identified (anatase, rutile and ilmenite). The anatase iso-form was predominant in the spectra obtained from smaller particles (68% + 32% rutile), suggesting these TiO_2_-NPs originate from paints and cosmetics. TEM imaging confirmed the presence of nanoscale Ti with smooth surface morphologies resembling engineered TiO_2_-NPs. 

There could be many implications of the TiO_2_-NPs accumulation in sludge, starting from the WWTP process itself. Zheng et al. [77] investigated the potential effects of TiO_2_-NPs on biological nitrogen and phosphorus removal and bacterial community in activated sludge. They found that concentration of 50 mg/L TiO_2_-NPs reduced the diversity of microbial community in the activated sludge and significantly decreased the removal efficiency of total nitrogen. In addition, Li et al. [78] reported that TiO_2_-NPs induce bacterial cell death modes which differentially weakened sludge dewatering. They also found that bacterial populations responses are more sensitive to the crystalline phases than dosage.

In a recent review on safety implications on the use of TiO_2_-NPs, Musial et al. [79] discussed the lack of consensus about the safety of this nanomaterial in the research community. Thus, there could even be a risk during the sludge management, which normally is applied dried as an amendment in agricultural fields. This, for instance, may include exposing farmers to dust inhalation. Incorporation of the sludge in the soil, with consequent accumulation of TiO_2_-NPs, could influence the microbial communities and the crops. While implications of the TiO_2_-NPs release in the environment are difficult to evaluate, assessing whether it is hazardous or not is beyond the scope of this study. However, this work has presented clear evidence of nanoscale Ti and its speciation in all WWTP sludge samples. These results provide an important basis for future research on the effects of sludge amendment on agricultural soils, soil microorganisms and plants.

## Figures and Tables

**Figure 1 nanomaterials-12-00744-f001:**
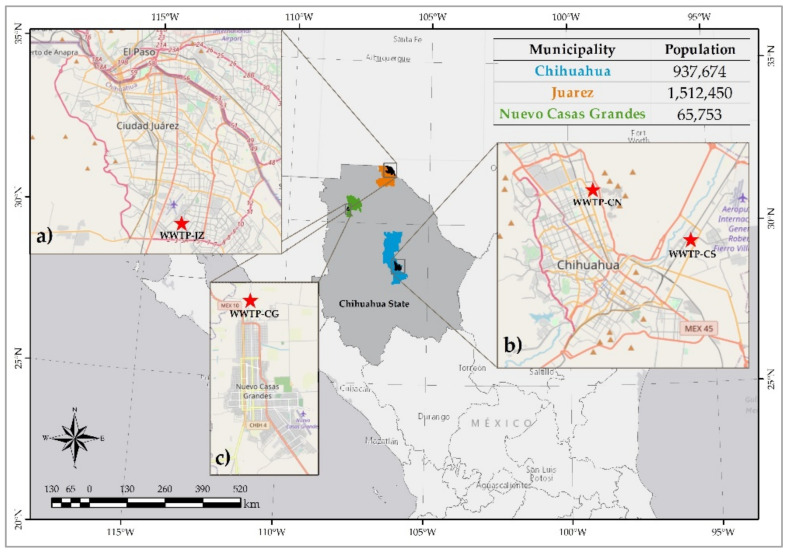
Localization of the WWTPs in the Chihuahua State, northern Mexico; the population size of each city is included in a table at the top-right of the figure.

**Figure 2 nanomaterials-12-00744-f002:**
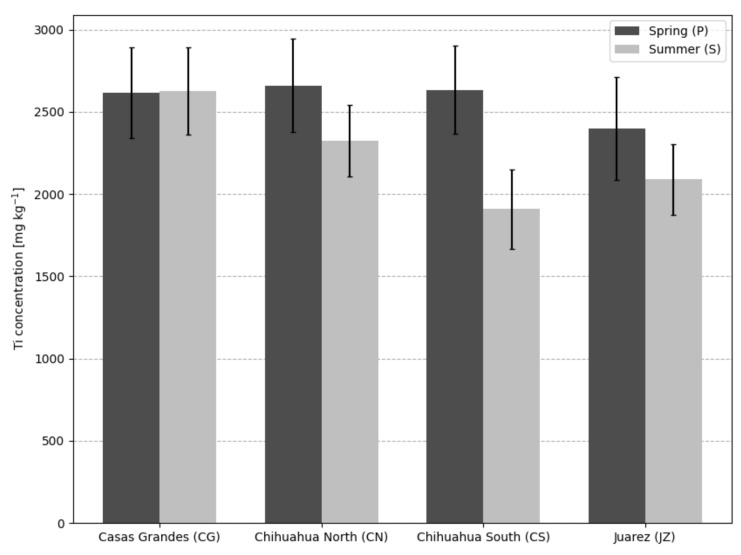
Average Ti concentration in sewage sludge by bulk XRF analysis. Error bars represent the combined standard uncertainty.

**Figure 3 nanomaterials-12-00744-f003:**
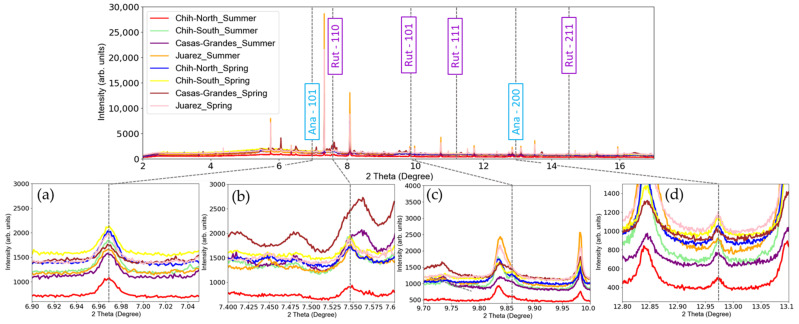
XRD diffraction spectra from all sludge samples. Zoomed regions are shown for the most important diffraction peaks of anatase (Ana) and rutile (Rut): (**a**) anatase 101; (**b**) rutile 110; (**c**) rutile 101 and (**d**) anatase 200. A small vertical shift has been applied for visualization purposes only.

**Figure 4 nanomaterials-12-00744-f004:**
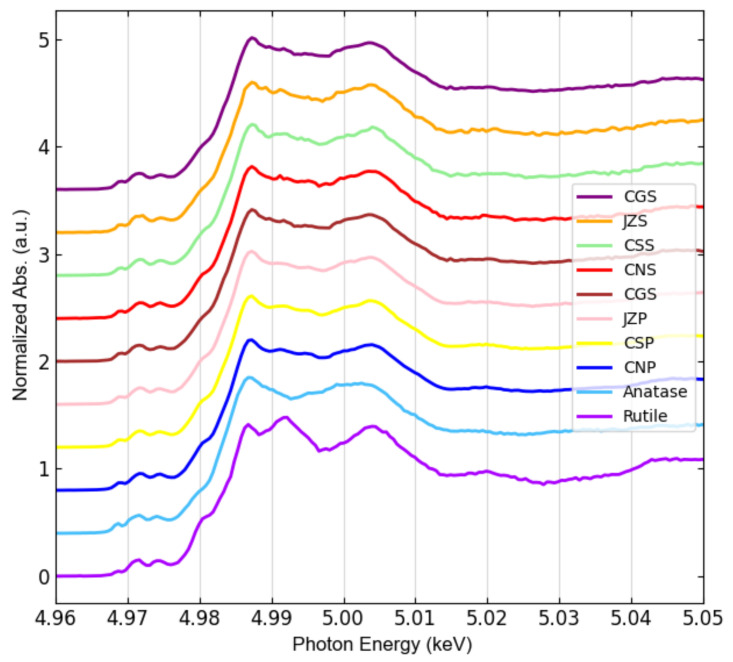
Bulk Ti K-edge XANES spectra of sludge samples, including rutile and anatase references. A vertical offset has been applied to help the comparison.

**Figure 5 nanomaterials-12-00744-f005:**
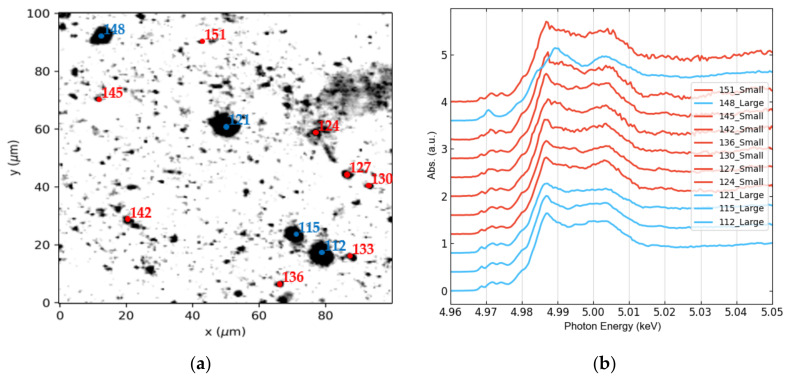
Example of Ti K-edge μXANES spectra acquired from hotspots of its total level distribution map from JZP: (**a**) μXRF map of Ti including some hotspots selected for μXANES analysis; (**b**) Corresponding XANES spectra. The color code for Small spots is red and for Large spots is blue.

**Figure 6 nanomaterials-12-00744-f006:**
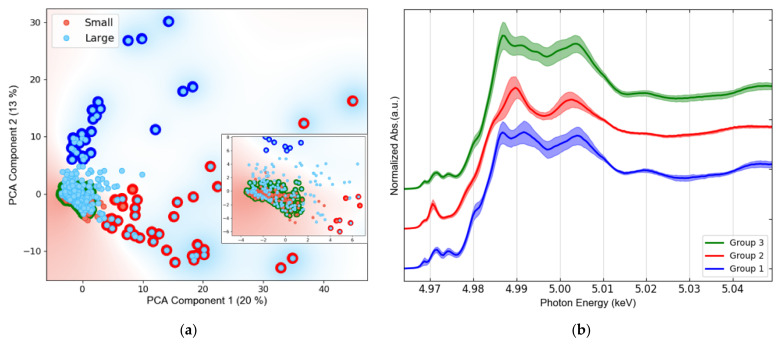
Results of PCA of all the Ti K-edge μXANES spectra: (**a**) PCA scores plot, including a zoom in the region with the highest point density. Between brackets, the percentage of total variance explained by each component is reported; (**b**) Corresponding average XANES spectra of each of the three identified groups (solid line) with uncertainty bands (standard deviation).

**Figure 7 nanomaterials-12-00744-f007:**
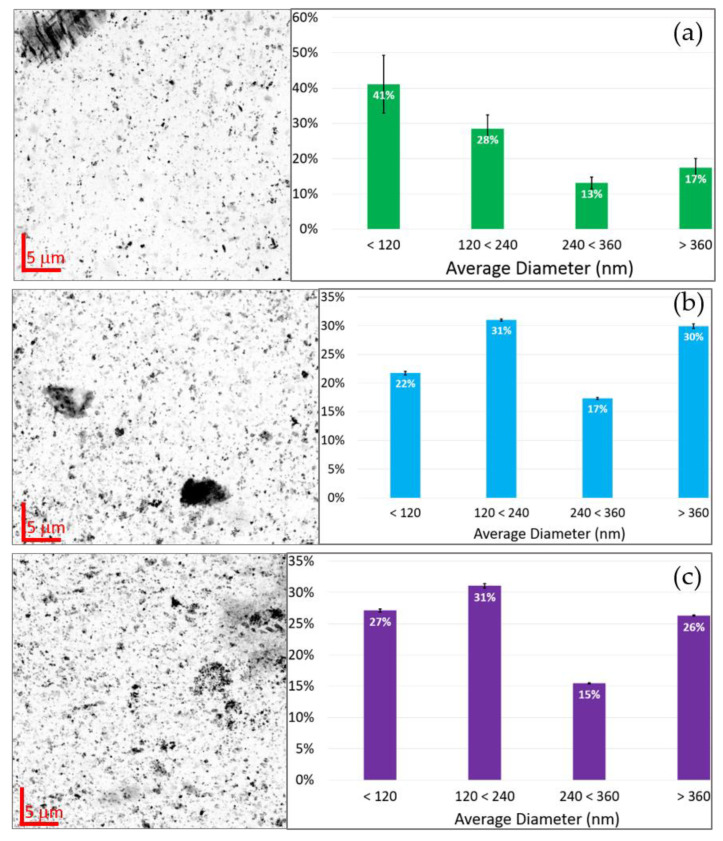
Titanium maps obtained from the nanoXRF analysis of sludge and corresponding particle diameter distribution: (**a**) Casas Grandes Spring (CGP); (**b**) Juarez Spring (JZP); and (**c**) Chihuahua North Spring (CNP).

**Figure 8 nanomaterials-12-00744-f008:**
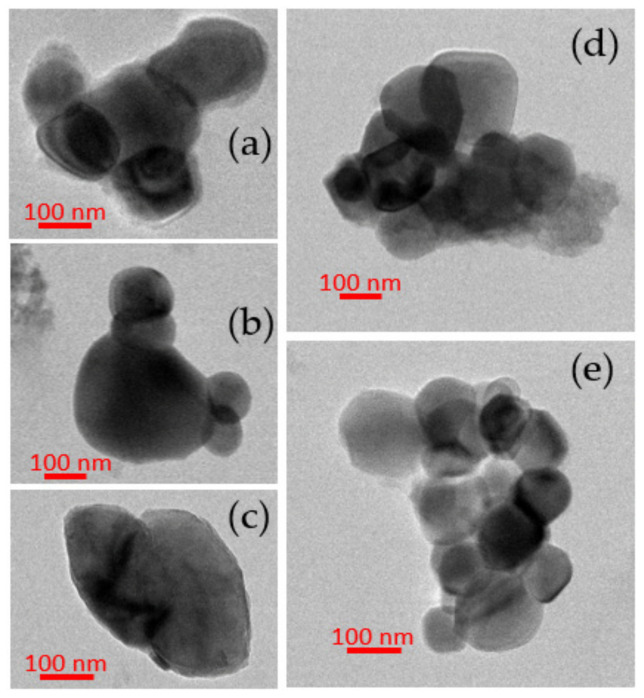
TEM images of particles detected in WWTP sludges: (**a**) Chihuahua South Summer; (**b**) Chihuahua South Spring; (**c**) Casas Grandes Spring; (**d**) Chihuahua North Spring; and (**e**) Juarez Summer.

**Figure 9 nanomaterials-12-00744-f009:**
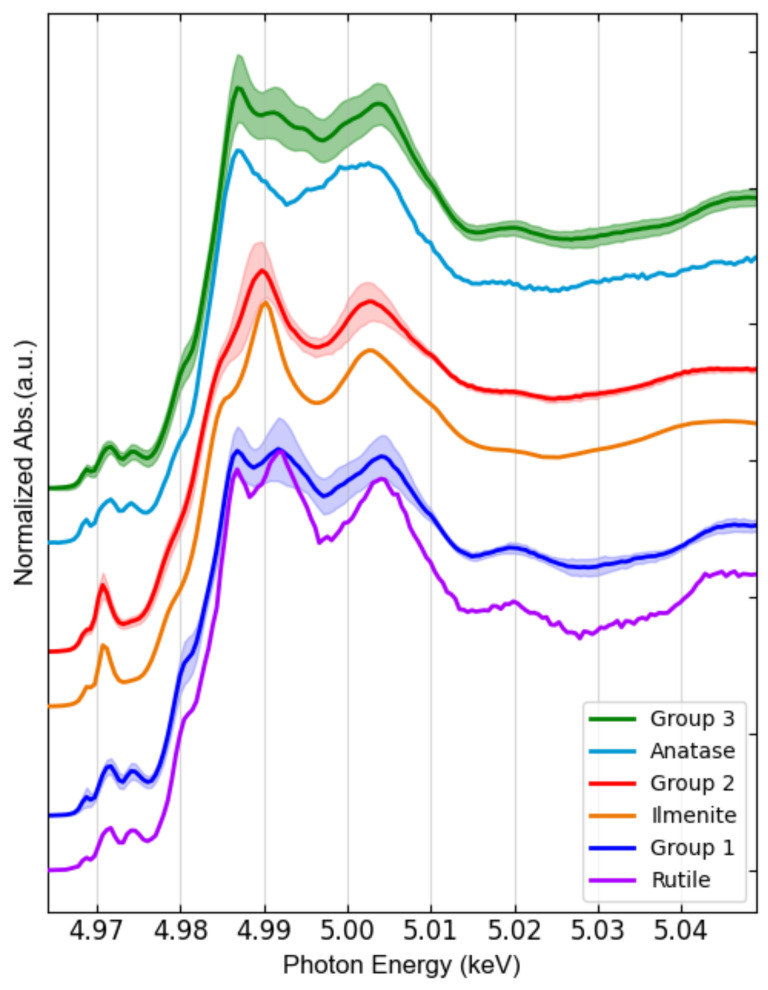
Comparison of the sludge Ti K-edge XANES spectra of three main groups of Ti hotspots identified by PCA, compared to references spectra of TiO_2_ phases. Anatase and rutile references were measured at ID21, ESRF. Ilmenite was obtained from XAS database of the beamline 10.3.2 at the Advanced Light Source, Berkeley, CA, USA.

**Table 1 nanomaterials-12-00744-t001:** The total concentration of trace elements [mg/kg] in the sewage sludge samples. The values are the average of two measurements. Combined standard uncertainty is around 10% for each measurement [58].

Element	CGP	CGS	CNP	CNS	CSP	CSS	JZP	JZS
Ti	2615	2625	2660	2325	2635	1910	2400	2090
Si	50,100	74,200	22,650	18,950	28,750	15,050	48,050	20,100
S	4370	3170	6190	3960	4850	3890	8665	25,200
Cl	1215	1130	1340	970	1395	873	1215	836
P	8450	1925	15,350	17,600	11,250	9275	13,750	14,750
K	6530	9840	2815	2310	2715	1500	3445	1065
Ca	23,850	18,000	41,700	35,700	39,150	24,300	56,400	38,950
Mn	158	216	143	115	144	98	176	150
Fe	13,350	17,050	7820	6095	7605	5095	25,300	22,250
Br	33	23	41	34	36	40	30	36
Ni	20	15	60	71	26	25	24	22
Cr	16	22	44	51	40	32	92	61
Cu	172	128	270	268	371	228	325	362
Zn	1135	741	1525	1405	1300	1017	2230	2365
Pb	165	114	141	114	147	126	132	123
Rb	62	72	24	20	31	20	32	19
Sr	190	144	333	277	214	157	495	503
Zr	192	206	192	156	179	156	180	188

**Table 2 nanomaterials-12-00744-t002:** Total Ti concentrations in WWTPs sludge from the literature, including this study.

Reference	WWTP Sites	Ti Concentration [mg/kg]
Choi et al. [31]	Maryland, US	1200 to 4670
Johnson et al. [68]	Southern region, England	370 to 670
Khosravi et al. [69]	Peterborough, Canada	320
Kim et al. [26]	Midwest and West regions of the USA	960 to 4510
Kiser et al. [18]	Central Arizona, USA	1100
Polesel et al. [30]	Trondheim, Norway	700
Shi et al. [28]	Shijiazhuang, Hebei Province, China	1360
Tong et al. [27]	Skokie, IL, USA	1700
Tou et al. [29]	Shanghai, China	33 to 2700
Wielinski et al. [32]	Wallisellen, Switzerland	1000 to 1800
This study	Chihuahua State, Mexico	1900 to 2600

## Data Availability

Data from microXANES spectra and its Orange workflow analysis can be found in the following link: https://github.com/jureyherrera/TiO2-NPs_sludge_WWTP_Chihuahua, accessed on 21 February 2022.

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
