# Peer review of "Detection and Characterization of TiO_2_ Nanomaterials in Sludge from Wastewater Treatment Plants of Chihuahua State, Mexico"

_nanomaterials, 2022, doi:10.3390/nano12050744_

Round 1

Reviewer 1 Report

The manuscript emphasized the investigation of TiO2 nanomaterial in the sludge from wastewater plant. I recommended a najor revision of manuscript. The corresponding comments are as follows.

  1. The distribution, concentration and transport should be considered in the abstract. It is noted that the common description of TiO2 nanomaterials was too much in the abstract. In addition, the keywords should be concise.
  2. I suggested that the section of 1.1 Nano TiO2 can be moved to the supplementary files. Instead, the transport of TiO2 nanoparticles in gas/water/soil environment should be introduced.
  3. “there are a few studies on characterization of ……”. But many references concern the TiO2 nanomaterials in wastewater plant, so that it can not be categorized as few studies. Thereby, the novelty of this investigation should be given.
  4. If possible, the introduction should be completely rewritten.
  5. If possible, the concentrations and distribution of TiO2 nanoparticles should be clearly competed in terms of the four different cities.
  6. It is important to notice that all anatase and rutile are from the used TiO2 nanomaterials, not formed in the transport and/or biological performance.
  7. The conclusion should be given.

Author Response

Dear reviewer, we really appreciate all your comments, which undoubtedly support the manuscript improvement. Below are the responses to each of your comments.

  1. The distribution, concentration and transport should be considered in the abstract. It is noted that the common description of TiO2 nanomaterials was too much in the abstract. In addition, the keywords should be concise.

Answer: The abstract was modified accordingly in order to better describe this research and main findings and conclusions. The keywords have been changed as well. 

  1. I suggested that the section of 1.1 Nano TiO2 can be moved to the supplementary files. Instead, the transport of TiO2 nanoparticles in gas/water/soil environment should be introduced.

Answer: It was decided to keep 1.1 section, nevertheless, considering your suggestion, a section denominated “1.2 TiO2-NPs release to the environment” was added, also extra information from line 52 to 55.

  1. “there are a few studies on characterization of ……”. But many references concern the TiO2 nanomaterials in wastewater plant, so that it can not be categorized as few studies. Thereby, the novelty of this investigation should be given.

Answer: Line has been changed (line 64),, besides in line 146  are explained the variety of techniques used in this research. This study offers novelty in the analytical approach by using several bulk and imaging techniques to characterize Ti in the sludge samples. In particular, the microXANES data is composed of more than 500 spectra and interpreted based on a multivariate analysis model. The groups obtained from this procedure have been then interpreted based on linear combination fitting. This number of spectra and analysis approaches have never been reported in another study. This research is also the first one from wastewater treatment plants from Mexico, and covers all plants from Chihuahua state (the largest state in Mexico). We have added a sentence to highlight this in line 150.

  1. If possible, the introduction should be completely rewritten.

Answer: The introduction was not completely rewritten, but considering your comments it was improved, also section “1.3. Fate of TiO2-NPs in WWTPs sludge” was restructured.

  1. If possible, the concentrations and distribution of TiO2 nanoparticles should be clearly competed in terms of the four different cities.

Answer: 

  1. Concentrations: Total Ti concentrations for the different sites have been included in table 1.
  2. Distributions: Due to the limited access to the nanoXRF beamline ID16B, the sub-set of samples was chosen based on the evidence that the cities and seasons do not present significant differences in Ti content, bulk-XRD and bulk-XANES speciation. Nevertheless, the size distribution has been obtained for the three concerned cities: Casas Grandes, Juarez and Chihuahua (please note that there are two WWTPs, North and South, in the Chihuahua city). 

  1. It is important to notice that all anatase and rutile are from the used TiO2 nanomaterials, not formed in the transport and/or biological performance.

Answer: Your concern was understood regarding the formation of the NPs in the transport or during the wastewater treatment process. However, it can’t assure this is not happening since the samples were recovered from operating wastewater treatment plants. Under this realistic scenario the aim of this research was to detect and characterize the Ti nanomaterials present in the sludge samples.Then as a secondary objective was identifying or narrowing down the source of these NPs. 

The data clearly reports the success in detecting TiO2-NPs in all samples, and then it was provided evidence that suggests anthropogenic origin based on the morphology and speciation:

Line 380: The µXANES results obtained in this study suggest that nano-anatase containing products are the main source of Ti. This isoform is used in pigments, indoor paintings and coatings due to its photocatalytic properties, as well as food additives [72]. Rutile was only a minor phase identified in all samples. This TiO2 isoform is mainly employed in UV light stabilizers and in sunscreens as a nanocomposite with sub-micrometric size [72]. 

Line 408:  However, natural Ti-NPs present rough surfaces, whereas engineered Ti-NPs present smoother surfaces, as it has been pointed out by Pradas et al. [24] based on TEM images. In this study, Ti-NPs mainly present smooth surfaces, which actually resemble engineered TiO2-NPs, such as P25 used in the study of Nickel et al. [75]. P25 are one of the most efficient commercial photo-catalyst TiO2-NPs, used in the industry of photo catalysis and in cosmetics, constituted by anatase (82%-86%) and rutile (18%-14%) [60, 63]. Furthermore, higher presence of anatase in sludge samples supports the hypothesis of its artificial origin [72] because in nature the most common minerals of Ti are rather ilmenite, rutile, brookite, and more rarely anatase [76]. 

  1. The conclusion should be given.

Answer: We agree that the conclusions section was not clearly identified and was rather dispersed in the discussion section. In order to address this issue we have added a new section “5. Conclusions and future perspectives”

Reviewer 2 Report

This study focuses on the physico-chemical characterization of titanium nanoparticles from sludges collected at various WWTPs in the city of Mexico. The study is well designed, and the protocol of investigation is clearly described. The authors should consider my comments here below for the improvement of the manuscript.

The abstract lacks a brief conclusion to highlight the implication of the study.

The Authors must clearly state the particularity of this study compared to previous related studies.

In section 2.1, the authors must mention the various seasons considered in this stydy.

What were the samples storage conditions?

XAS must be written in full for the first time it appears in the text.

The authors must amend the title of the Y-axis in Figure 6(b).

The authors must abstain from using the personal pronoun: e.g. In this study we observed… Line 384.

Author Response

Dear reviewer, we really appreciate all your comments, which undoubtedly support the manuscript improvement. Below are the responses to each of your comments.

1. The abstract lacks a brief conclusion to highlight the implication of the study. 
Answer: We agree that the conclusions section was not clearly identified and was rather dispersed in the discussion section. In order to address this issue we have added a new section “5. Conclusions and future perspectives”

2. The Authors must clearly state the particularity of this study compared to previous related studies.
Answer: This study offers novelty in the analytical approach by using several bulk and imaging techniques to characterize Ti in the sludge samples, defined in line 146. In particular, the microXANES data is composed of more than 500 spectra and interpreted based on a multivariate analysis model. The groups obtained from this procedure have been then interpreted based on linear combination fitting. This number of spectra and analysis approaches have never been reported in another study. This research is also the first one from wastewater treatment plants from Mexico, and covers all plants from Chihuahua state (the largest state in Mexico). We have added a sentence to highlight this in line 150.

3. In section 2.1, the authors must mention the various seasons considered in this study.
Answer: This has been stated in the abstract and just at the beginning of section 2.1: During spring (P) and summer (S) of 2017 line 154. 

4. What were the samples storage conditions?
Answer: The sludge was dried at ambient temperature and stored in plastic bags with hermetic closure, this information was added in line 166.

5. XAS must be written in full for the first time it appears in the text. 
Answer: FXAS was defined and the first time mentioned is in line 208.

6. The authors must amend the title of the Y-axis in Figure 6(b). 
Answer: Figure 6(b) and Figure 9 Y-axis labels have been corrected.

7. The authors must abstain from using the personal pronoun: e.g. In this study we observed… Line 384. 
Answer: This suggestion has been changed in line 408.

Reviewer 3 Report

Dear Editor

The study describes the determination and characterization of TiO2 in sludge, the novelty of the study is Ok. The characterization and discussion are designed well.

Hence, the following comments need to be addressed.

1-Title is not clear; determination of TiO2 ? Removal of TiO2? Extraction of TiO2? Please clear it.

2- Abstract is general. “Used nanomaterials worldwide” which nanoparticles? TiO2? In abstract should discuss the aim and scope of the current study. Also: “their re-activity, toxicity and therefore their environmental effects.”  Have you studied the toxicity of TO2? if no, change and rephrase it.

3- Introduction Line 38: “including food,” add example.

4- Line 96: “ … indicated TiO2-NPs 96 as a potential carcinogen…”. Is there a maximum permission level for TiO2?

5- Ti is missing in Table 1. It's better to add Ti in Table 1 for better comparison.

6- From TEM images, how do you are detected that the appeared particles are TiO2? Are phases studied? d-spacing?

Author Response

Dear reviewer, we really appreciate all your comments, which undoubtedly support the manuscript improvement. Below are the responses to each of your comments.

The study describes the determination and characterization of TiO2 in sludge, the novelty of the study is Ok. The characterization and discussion are designed well.
Hence, the following comments need to be addressed.

1. Title is not clear; determination of TiO2 ? Removal of TiO2? Extraction of TiO2? Please clear it.   
Answer: The words suggested as removal or extraction do not apply to the objective of this research. Considering your comment, It was decided just to be more specific with the title “Detection and characterization of TiO2 nanomaterials in sludge from wastewater treatment plants of Chihuahua State, Mexico”

2. Abstract is general. “Used nanomaterials worldwide” which nanoparticles? TiO2? In abstract should discuss the aim and scope of the current study. Also: “their re-activity, toxicity and therefore their environmental effects.”  Have you studied the toxicity of TO2? if no, change and rephrase it.
Answer: The abstract has been modified in order to address this issue.

3. Introduction Line 38: “including food,” add example.
Answer: an example has been added in line 40, food-grade TiO2/E171, which reference is there (Weir et al. 2012 [6])

4. Line 96: “ … indicated TiO2-NPs 96 as a potential carcinogen…”. Is there a maximum permission level for TiO2?
Answer: "NIOSH recommended exposure limits of 2.4 mg/m3 for fine TiO2 and 0.3 mg/m3 for ultrafine (including engineered nanoscale) TiO2, as time-weighted average (TWA) concentrations for up to 10 hours per day during a 40-hour work week." This information was added on the introduction in line 109.

5. Ti is missing in Table 1. It's better to add Ti in Table 1 for better comparison.
Answer: Ti concentrations have been added in Table 1.

6. From TEM images, how do you are detected that the appeared particles are TiO2? Are phases studied? d-spacing?
Answer: During TEM imaging, titanium presence in the NPs has been done by EDX analysis, where Ti and O were the only elements contained on the NPs images (Figure A4). There aren’t measurements of phases, for example by electron diffraction, because of this, we have done some changes in order to refer only as Ti-NPs regarding the TEM imaging in line 314.

Round 2

Reviewer 1 Report

The manuscript was revised in accordance with the reviewer’s suggestions. I recommended an acceptance of the manuscript.

Reviewer 2 Report

Satisfied with the amended paper.